# Towards Intensive Co-operated Agribusiness: A Gender-Based Comparative Borich Needs Assessment Model Analysis of Beef Cattle Farmers in Eswatini

**Sicelo Ignatius Dlamini [1] and Wen-Chi Huang [2,*]**

1   Department of Tropical Agriculture and International Cooperation, National Pingtung University of Science and Technology, Pingtung 91201, Taiwan; revsicelodlamini@gmail.com
2   Department of Agribusiness Management, National Pingtung University of Science and Technology, Pingtung 91201, Taiwan
*   Correspondence: wenchi@mail.npust.edu.tw; Tel.: +886-8-7703203 (ext. 7824); Fax: +886-8-7740190

**Abstract:** Beef cattle farming assumes a pivotal role in economic growth, household food security, and poverty alleviation in Eswatini. However, paucity of information dissemination, and competence are drawbacks that accord a steady annual increase in beef imports and a decline in exports. Therefore, the study conducted a gender-based comparative assessment of training needs for beef cattle farmers. Primary data were collected through personal interviews, guided by a reliability-tested questionnaire, from a sample of 397 farmers. The Borich Needs Assessment Model was adopted for data analysis and inferential statistics were employed to evaluate statistically significant differences between the gender groups. On a scale of 5, farmers were found to be less proficient ($M = 1.891$, $SD = 0.529$) in cattle production and agribusiness management practices. Female farmers were significantly less proficient than males ($t = -6.004$, $p = 0.000$). Statistically significant differences in mean weighted discrepancy scores ($t = 5.280$, $p = 0.000$) revealed a strong training need for females compared to men. It is recommended that dissemination of training information should be prioritized as follows: (1) agribusiness management concepts, (2) feed and feeding concepts, (3) cattle health concepts, (4) farmer-organizational concepts, (5) farm structures, and (6) breeding and rearing concepts.

**Keywords:** Borich needs assessment model; proficiency; beef cattle farmers; *t*-test; Eswatini

## 1. Introduction

Since time immemorial, beef cattle farming has been at the center of the lives of the people of Eswatini. The people keep beef cattle for subsistence and agribusiness purposes, although the latter is to a lesser extent. Traditionally, cattle play a pivotal role in Swati rituals and social status, thus venerated for their high economic and social value. This has elevated the livestock husbandry to be the second most vital subsector of agriculture. Globally, livestock husbandry dispenses almost half of the agricultural global output [1]. The rapid food revolution evident in the increase in the demand for livestock products, especially in developing countries, anchors the livestock husbandry at the center of rural livelihood advancement and poverty alleviation.

Developing economies, such as Eswatini where livestock production is popular, ought to take an unusual business approach to seize domestic and export business opportunities through the livestock husbandry. Taking advantage of available agribusiness opportunities requires a farmer-motivated shift from subsistence to commercial farming. For beef cattle farming in Eswatini, this can be achieved through intensification in the production and marketing systems. Given the lack of government funds

for rural development in Africa as a whole, where donor-funded agricultural development projects often collapse, demand-based capacitation and empowerment on production systems and resource mobilization strategies is deemed a sustainable mechanism for promoting agribusiness-oriented agriculture. Hence, the exigency of farmer-evaluated training needs assessment required for the shift towards intensive agribusiness for the amelioration of rural livelihoods.

The domestic and export shortfalls in beef supply, in Eswatini, are indicative of the potential of the beef cattle agribusiness enterprise as a prospective avenue for rural and national economic growth. Beef imports for domestic consumption have been steadily increasing on an annual basis [2]. Recent statistics reflect that beef imports amounted to 2513.08 tons, and a further decline in beef exports from 703.25 tons in 2016 to 27.62 tons in 2018 was noted [3]. This reveals a potential increase in domestic and export demand for beef, which is an agribusiness prospect for beef cattle farmers. Therefore, beef cattle production intensification is a potential strategy for the enhancement of rural livelihoods and the national economy. Farmers can seize the available agribusiness opportunity to improve their livelihoods.

However, the adoption and implementation of any agribusiness development strategy requires farmers to exhibit competitive levels of knowledge and skills [4]. Farmers ought to expand their knowledge and skills in production, networking, marketing, organization, and mobilization of resources. Thus, educational capacitation is an integral strategic issue in rural development and livelihood improvement. In this regard, substantial training needs assessment research plays a critical role in addressing farmers' knowledge, needs and desires to prevent poor adoption of livelihood enhancement strategies and negative social consequences [5,6]. Withal, incorporating gender as a control variable in needs assessment research is useful in arresting gender-based competence discrepancies, which are often a source of poverty injustice and inequality in developing countries. Additionally, such a comparative research approach contributes to the literature that bridges gender-based poverty in rural communities.

It is, however, unfortunate that livestock production extension and information dissemination in livestock husbandry are rarely a priority in developing countries [6]. Development strategies, in general, largely exclude training and training needs assessment [7,8]. If in any case, development training programs often bypass the basic step of needs assessment research, effectuating poor adoption of new farming practices and management systems.

In Eswatini, beef cattle farmers rely on their traditional know-how for cattle production and marketing, since extension support is only offered in disease control through consultation. Information dissemination on beef cattle production and agribusiness management is scanty and there is no training needs assessment research that has been conducted. Needs assessment research is essential in identifying training needs and prioritizing information dissemination. Therefore, the primary objective of this study was to conduct a gender-based comparative assessment of training needs for beef cattle farmers, by assessing competency discrepancies in production and agribusiness-related management practices. Specifically, the study sought to:

I.      Describe the differences in self-evaluated levels of importance of competencies between male and female beef cattle farmers;

II.     Identify proficiency differences in competencies between the gender groups; and

III.    Evaluate the inter-gender group discrepancies in beef cattle production and related agribusiness management competencies among farmers.

## 2. Literature Review

### 2.1. Farmer Training in Eswatini

Farmer training is the primary responsibility of the Department of Agricultural and Extension Services of the Ministry of Agriculture [9]. The department is mandated to provide programs and activities on sustainable agriculture production to build farmers' capacity through extension services.

Specifically, the training function is offered by the Extension Services Section, whose priority objective is to equip farmers with relevant skills for increased agricultural production.

In recent years, the lack of funds in government forced the department to withdraw extension officers that were deployed in rural areas, which has left farmers without immediate extension support. As it were, the extension officers were mainly deployed for crop production support, relegating livestock production support to the Livestock Services Section of the Department of Veterinary and Livestock Services. This created a gap in livestock production and management since the Livestock Service Section focuses on veterinary and marketing services. Although the department's mandate covers the provision of adequate knowledge, skills and technical expertise, a lot is left to be desired in this regard.

As in other countries, [10], lack of scientific knowledge and skills in cattle production is a major challenge in Eswatini. Training courses on livestock production are becoming popular with input suppliers, at a cost that is not necessarily friendly to poor rural farmers. The government's focus on cash crop farming, such as sugarcane, continues to impose neglect on livestock farmers' training needs, especially beef cattle farmers. Therefore, this study is critical in providing foundational literature that forms the basis for the development of accurate and comprehensive training programs for beef cattle farmers.

In view of the importance of beef cattle farming in the agriculture subsector and national economy, Eswatini cannot afford to leave farmer training in the hands of a traditional informal education system. Programs must be mounted to ensure adequate knowledge, skills, and technical expertise that are vital for rural food security and advancement of livelihoods. The current study provides a strategic tool for initiating the processes of program development and information dissemination to beef cattle farmers.

## 2.2. Training Needs Assessment

From an agriculture perspective, training is the impartation of agricultural knowledge and skill to capacitate the human capital involved in production, management, and marketing processes. Training is an effective tool for implementing development programs and inducing positive change [11]. It aims at imparting the desired knowledge, skills, and competencies to improve the performance of farmers [12,13]. Farmer training also works well in addressing misconceptions on production, health, and management practices [11]. On the other hand, a need is an intrinsic drive that thrusts farmers towards improved performance of a task [14].

Ideally, educational empowerment requires the identification of the discrepancy between "what the farmers should know" and "what they currently know," to establish the necessary congruence through accurate extension services. This discrepancy is termed "a training need" [15–17]. The actual exercise of identifying and ranking training needs is referred to as a "training need assessment." Practically, the "what should be" polar involves measurable behaviors often called "skills", whereas the "what is" position is evident in the efficient and effective way in which the farmers execute acquired skills [17]. Therefore, the distance between these poles serves as an index used for ranking the required training needs. Positive indices with high values connote priority in the rank order of the training needs.

The imperative of addressing a training need is in improving the farmer's ability to perform a task effectively and efficiently [18]. Training needs assessment serves as a strategic planning issue essential in addressing not only the discrepancies but also the attitudes of the agents involved, the trainer and trainee. Trainers, develop evidence-based capacitation programs, thus eliminating the often, non-successful top-down approach towards problem-solving. On the other hand, the trainees learn to integrate their experiences with new technologies, management programs, and solutions. Hence, needs assessment-based training stimulates an intrinsic force for change towards seeking capacity for performance reforms in production and management.

In addition, training needs assessment is critical in establishing the relevant strategies of information dissemination such as workshops, seminars, and farmer field schools [19,20]. It also

reveals the availability and interest of farmers to undertake training, identifying appropriate time slots for training [10]. This transforms training needs assessment into an indispensable component in rural development strategy, ensuring smooth and effective facilitation and adoption of new and/or improved farming practices and systems. In the same vein, needs assessment research is critical in cases where cognitive shift is a priority, such as the shift from subsistence to commercial farming [21]. Beef cattle farming in Eswatini exhibits low market participation and near-zero market orientation, with farmers selling old-aged cattle under distress sales for immediate cash needs. Advancing the idea of income generation over wealth storage requires careful introduction of agribusiness concepts based on needs assessment. Hence, this study builds a farmer-intrinsic desire to venture into agribusiness, otherwise, top-down approaches would result in conflicts with the traditional belief in subsistence cattle farming.

Generally, most of the models of program design manifest a similarity in analyzing the context and recipients to identify the problem [22]. An assessment specifically identifies farmers' problems, needs and strengths [23] for action, to develop farmers-centric or demand-led programs that address existing real problems on the field [24,25]. Otherwise, any deviation from consultative communication networks systems, such as needs assessments, leads to poor quality outcomes [26] and has relegated agriculture to be less remunerative [27]. Such variance further stifles the fight against food insecurity, poverty, and inequality among African states, defeating the ultimate utility of training programs as it propagates ambiguous training programs [28]. Unambiguity in program design and information dissemination is achieved through specificity in training needs assessment research, focusing on distinct enterprises within a subsector. Thus, there is a dire need to embark on needs assessment to capture the proficiency deviations among beef cattle farmers in Eswatini.

*2.3. Gender-Based Training Needs Assessment*

Meanwhile, various researchers have embraced a focus on special target groups, such as women in agriculture [29–32]. Rural women command a strong desire to engage in household agricultural income-generating activities [33]. Female farmers are involved in both crop and livestock husbandries. In Eswatini, recent statistics indicate that women account for 11% of the agriculture labor force [34]. In rural agriculture, women are often left with the responsibility of farming and livestock rearing, as male household heads attend to off-farm employment. Furthermore, local traditional agricultural markets are mainly served by women, revealing the importance of women in both production and marketing functions.

Inasmuch as women are central in rural agriculture, it is an open secret, especially in developing countries, that female farmers encounter a unique set of challenges compared to their male counterparts. Commonly in Africa, women still lack the right to resource acquisition [35], such as land, due to some rural traditional and cultural customs. Although it has been a recent constitutional right, it is still difficult for a woman to secure land on Eswatini Nation Land.

More often than not, women labor productivity is comparatively lower than men due to social and economic constraints [36]. Female farmers are often less educated and less privy to training opportunities and production resources [37]. They lack technical and scientific knowledge on livestock production [38]. A study by Barbercheck, Brasier, Kiernan, Sachs, Trauger, Findeis, Stone, and Moist [30] further found that women farmers are not taken seriously by extension offers, compared to their male counterparts. In Eswatini, the lack of knowledge is also exacerbated by the traditional notion that women are best suitable for small livestock production and management. This has attached women to livestock such as chickens and goats, depriving them of household training in larger livestock such as beef cattle rearing, yet they assume the right to household cattle ownership at the demise of the male household head. This has led to the collapse of several household cattle enterprises, threatening rural food security. Hence, the development of training frameworks that encapsulate female farmers' special needs is crucial in the fight against food insecurity, poverty, and inequality [26].

## 3. Materials and Methods

### 3.1. Study Area

The study area covers a total land area of 17,364 km$^2$, under a dual land tenure system; Eswatini Nation Land (ENL) and Title Deed Land (TDL). ENL is land held in trust by the king for the nation, whereas TDL is land owned by individuals or companies [39]. The country is agrarian, with about 70% of the population dependent on subsistence agriculture in rural areas. The poverty rate is estimated at 63% of the population [40], and the unemployed rate at 41.8% [41].

The country's 1.2 million population [42] is distributed over the Hhohho, Lubombo, Manzini, and Shiselweni districts, through which beef cattle farmers are organized and managed by regional livestock officers. A veterinary assistant is deployed to manage four dip-tanks, conducting general disease surveillance and ensuring prophylactic control of ticks and tick-borne diseases [3]. Apart from private dip-tanks on TDL, each government-aided dip-tank serves ENL smallholder farmers within a radius of 7.5 km.

The study area is further divided into four ecological zones (Highveld, Middleveld, Lowveld, and Lubombo Plateau), based on the prevailing climatic conditions and the terrain. The climate varies from subtropical to near temperate over the ecological zones. The Highveld is characterized by a cool-wet climate, with an annual rainfall distribution between 700 mm and 1550 mm, while the Middleveld and Lubombo Plateau are warmer with an annual rainfall amount of 550 to 850 mm [43]. The Lowveld is hotter and dryer, receiving about 400 to 550 mm of rain. The high amounts of rainfall in the Highveld promotes soil acidity and varied grass types, reducing grazing quality. Hence Highveld cattle are of poor condition, although the region is not often affected by drought. The soils in the Middleveld and Lubombo Plateau are fertile, due to the good rainfall amounts, thus high-quality grazing and cattle condition. The Lowveld is composed of highly palatable grasses but often ravaged by recurrent drought, thus reducing the region's suitability for beef cattle farming.

### 3.2. Sampling and Data Collection

The study was conducted in all the four districts of Eswatini, targeting smallholder farmers on Eswatini Nation Land ($N$ = 50,985). The Slovin's formula [44] was applied to determine the sample size ($S$ = 397) as follows:

$$S = \frac{N}{1 + Ne^2} = \frac{50,985}{1 + 50,985(0.05)^2} = 396.886 \approx 397 \tag{1}$$

where: $S$ = desired sample size; $N$ = population of farmers; e: level of error tolerance (0.05).

The study adopted a multi-stage stratified random sampling technique to extract the sample. Farmers were first grouped into the four districts. Second, farmers were stratified according to gender, male and female, ensuring that both groups were well represented for the comparative assessment. The sample size for male farmers was denoted as $s_{male}$ = *199*, while $s_{female}$ = *198* represented the group of female farmers. The third stage involved simple random sampling to draw the samples from each stratum.

Personal interviews, guided by a structured questionnaire, were used to collect data from farmers from September to December 2018. The study focused on the core livestock production and agribusiness management competencies. A total of thirty-four (34) competency statements were developed in these aspects, out of which eighteen (18) related to cattle production and management practices. Production management practices are salient in maximization of the production of marketable cattle required for engagement in agribusiness. The competency items were categorized into breeding and rearing [32], pasture and grazing management [45], fodder production and storage [46], cattle nutritional needs and deficiencies [47], disease control [48], and construction and maintenance of cattle sheds and pens.

In order to cultivate a shift towards agribusiness-oriented agriculture, ten (10) additional competency statements related to agribusiness were developed and grouped into farm business

management, and marketing and pricing of beef cattle [49]. Agribusiness management practices are paramount in the organization of production efforts and the transaction of marketable surplus. Knowledge and skills in agribusiness and cattle marketing and pricing are key in enhancing farmers' competitiveness in the unorganized market structure in Eswatini [50]. This is vital in increasing the economic benefit that creates interest in agribusiness farming.

Since the lack of capital often stonewalls agribusiness development initiatives, cooperation among farmers is an integral framework for the mobilization and reorganization of resources for economic and social stability [51–53]. Hence, further six (6) competency statements related to the establishment and management of farmers' cooperatives were developed.

The questionnaire used a dual-five-point Likert-scale, minimizing confusion and respondent fatigue, to measure the self-evaluated levels of importance and proficiency of competency items. The Likert-scale was designated as follows: 1 = Very Low, 2 = Low, 3 = Intermediate, 4 = High, and 5 = Very High, for both importance and proficiency statements. Socio-economic characteristics were also collected, and a comprehensive gender comparison is presented in Table 1.

Before the data collection process, Cronbach's alpha ($\alpha$) measured from thirty (30) farmers was used to ascertain the questionnaire reliability using SPSS Version 22 (IBM, New York, U.S.A.). Cronbach's alpha assesses the internal consistency of dichotomous or multi-pointed formatted questionnaire items. The questionnaire indicated acceptable alpha coefficients, $\alpha = 0.845$ for importance and $\alpha = 0.868$ for proficiency.

**Table 1.** Gender-based comparison of farmers' socio-economic characteristics.

| Continuous Variable | Male ($s_{male} = 199$) | | Female ($s_{female} = 198$) | | *t*-Value |
|---|---|---|---|---|---|
| | Mean | SD | Mean | SD | |
| Age (years) | 56.52 | 13.401 | 60.25 | 12.000 | 2.921 *** |
| Education (years) | 9.88 | 4.385 | 8.00 | 4.507 | −4.222 *** |
| Herd size (number) | 20.97 | 15.254 | 11.31 | 9.892 | −7.491 *** |
| Experience (years) | 20.236 | 12.025 | 18.417 | 11.974 | −1.511 |
| **Categorical Variable** | **Male ($s_{male} = 199$)** | | **Female ($s_{female} = 198$)** | | $\chi^2$ |
| Farm location | Hhohho = 49, Lubombo = 50, Manzini = 50, Shiselweni = 50 | | Hhohho = 49, Lubombo = 50 Manzini = 49, Shiselweni = 50 | | 2.849 |
| Off-farm income (Emalangeni) | < E1000 = 57, E1000–E10 00 = 91, > E10 000 = 51 | | < E1000 = 84, E1000–E10 00 = 77, > E10 000 = 37 | | 8.562 ** |
| Marital Status | Single = 10, Married = 173, Widowed = 16 | | Single = 2, Married = 60, Widowed = 136 | | 154.871 *** |
| Ecological zone | Lowveld = 42, Middleveld = 115, Highveld = 42 | | Lowveld = 32, Middleveld = 109, Highveld = 57 | | 3.782 |

*p*-value in parentheses; ** $p < 0.05$; *** Significant at $p < 0.01$.

### 3.3. Analytical Framework

Training needs were evaluated using the self-evaluative Borich Needs Assessment Model (B-NAM). The underlying assumption of the B-NAM is that a farmer can objectively judge his or her ability in applying performance skills [54]. Proposed by Borich [16], the model is extensively utilized in agricultural education and extension to identify and evaluate training needs for agriculture teachers, extension officers, and farmers [5]. The model has merit for its ability to capture data based on both the respondent's present state and desired state of affairs. Ideally, the model measures the distance (discrepancy) between importance and proficiency in competency items as reported by the farmers. Weighted discrepancy scores (WDSs) are then computed to represent indices for training needs. The WDSs for this study were computed as follows [17]:

$$DS_{ij} = I_{ij} - P_{ij} \tag{2}$$

$$WDS_{ij} = DS_{ij} \times \bar{I}_j \tag{3}$$



$$MWDS_j = \sum WDS_{ij}/S \tag{4}$$

where: $DS$ = discrepancy score; $I$ = importance score; $P$ = proficiency score; $i$ = beef cattle farmer; $j$ = competency item; $WDS$ = weighted discrepancy score; $\bar{I}_j$ = mean importance score; $MWDS$ = mean weighted discrepancy score; $S$ = sample size.

Mean discrepancy scores were calculated using Excel, while SPSS Version 22 was utilized for running descriptive statistics (mean and standard deviation) and inferential statistics (independent *t*-test and chi-square test). Inferential statistics from sample data allow for the generalization of findings to a population. The *t*-test was suited for this study since it is a special form of Analysis of Variance for comparing mean scores between two groups. Pre-analysis descriptive analyses did not reveal serious violation of the *t*-test assumptions. Data normalization was further done to ensure that all variables were near normally distributed. This was further enhanced by the large enough sample size, $S = 397$ ($s_{male} = 199$, $s_{female} = 198$), which warranted the use of the independent *t*-test for this study [55–57]. The independent test was used to assess the significant differences between male and female farmers in relation to the continuous socio-economic characteristics (farmers' age, education, herd size, and experience), importance, proficiency, and mean weighted discrepancy scores. Statistical differences between gender groups based on categorical socio-economic characteristic (farm location, off-farm income, marital status and ecological zone) of the farmer were evaluated using the chi-square tests. For interpretation purposes of the test statistics, alpha ($\alpha$) was set at $p < 0.05$.

Forbye, effect size was utilized as a supplementary statistic to validate the independent *t*-test [58]. Effect size is an effect statistic or index [59] for measuring the magnitude of the difference between two group means [60]. Evaluating the size of the effect of the independent variable on a dependent variable is essential in prioritizing follow-up action [61]. Common in literature, the effect size associated with the parametric Student's *t*-test, Cohen's d, was adopted and computed as follows:

$$\text{Cohen's d} = \frac{M_{Male} - M_{Female}}{SD_{pooled}} \tag{5}$$

where; Cohen's d = effect size; $M_{Male}$ = male group mean; $M_{Female}$ = female group mean; $SD_{pooled}$ was computed as:

$$SD_{pooled} = \sqrt{(SD^2_{Male} + SD^2_{Female})/2} \tag{6}$$

where: $SD^2_{Male}$ = squared standard deviation of the male group; $SD^2_{Female}$ = squared standard deviation of the female group. For interpretation purposes, $d < 0.50$ indicated small effect size; $0.50 \geq d < 0.80$ indicated moderate effect size and $0.80 \leq d$ reflected large effect size [62]. Pooling of the variances produces better estimates of the assumed equal variances between the two groups [55].

In order to confirm the association between the control variable, gender, and the importance and proficiency of farmers, the point-biserial correlation analysis was performed. Point-biserial correlation is a special instance of the Pearson correlation used to assess the strength of association between a dichotomous independent variable and a continuous dependent variable [63].

## 4. Results and Discussion

### 4.1. Descriptive Statistics

The study purpose was to conduct a gender-based comparative assessment of training needs for beef cattle farmers. Table 1 reveals significant differences between the gender groups with respect to some socio-economic characteristics (farmer's age, education, herd size, off-farm income, and marital status). These variables have the potential of influencing farmers' proficiencies and training needs. Farmer's age has a positive and/or negative effect on production performance. The significantly higher average age for females ($M_{Male} = 56.52$, $M_{Female} = 60.25$, $t = 2.921$, $p = 0.004$) represents a negative effect on proficiency since females only assume the traditional right to household cattle ownership after the

demise of the male household head. This means that even though the female farmer is older, she has less competence in the production and agribusiness related processes.

Differences in education levels reflect differences in proficiency levels between the gender-groups. The results indicate higher significant average schooling years for males compared to females ($M_{Male}$ = 9.88, $M_{Female}$ = 8.00, $t$ = −4.222, $p$ = 0.000). This directly means that males are significantly more educated than females, implying that females are expected to have a greater training need than males. A study by Rais, Solangi, and Sahito [35] found rural women folk to be more illiterate than the male folk, requiring more agricultural training than their male counterparts. Significantly higher herd size for males than females ($M_{Male}$ = 20.97, $M_{Female}$ = 11.31, $t$ = −1.511, $p$ = 0.000) implies that males have higher proficiency in beef cattle production compared to their female counterparts, suggesting a greater training need for females. Off-farm income reflects the availability of capital that could be re-invested into beef cattle farming. The variable is found to be statistically significant ($p$ = 0.014), indicating that males tend to have more off-farm income than females. Directly, this means males are better able to solicit inputs than females, implying that males are expected to be more productive than females. Marital status reflects the availability of labor and family income. The results indicate that more males are married than females ($Males_{Married}$ = 173, $Females_{Married}$ = 60) and fewer males are widowed than females ($Males_{Widowed}$ = 16, $Females_{Widowed}$ = 136). The finding reveals a wider labor and management base for males than females. This also reflects that males command higher production and decision-making capacity than females, thus there is a greater training need for females than males.

The point-biserial correlation (see Table 2) confirms a strong significant association ($p$ < 0.01) between gender and proficiency ($M$ = 1.891, $SD$ = 0.529, $r_{pb}$ = 0.289) and the mean weighted discrepancy scores ($M$ = 11.692, $SD$ = 2.433, $r_{pb}$ = −0.257). Further significant association ($p$ < 0.01) is observed between importance, and proficiency and mean weighted discrepancy scores, $r_{pb}$ = 0.234, $r_{pb}$ = 0.316, respectively. Although the correlation coefficients are small, the strong level of significance reveals the importance of the associations in the wider population. Moreover, when interpreted as Cohens' d effect size, the correlations > 0.3 represent a medium magnitude of association [63], thus, providing a substantive basis for our study design. The correlation between proficiency and the mean weighted discrepancy scores reveals a strong negative association between the two assessed attributes, $r_{pb}$ = −0.849 at $p$ < 0.01. The negative sign captures the fact that that MWDS tend to be lower with males (males = 1, females = 0), implying that training need is lower for males than females. These results ascertain the justification for the gender comparative assessment design of this study.

**Table 2.** Point-biserial correlation ($S$ = 397).

|  | **Mean** | **SD** | **Gender** | **Importance** | **Proficiency** | **MWDS** |
|---|---|---|---|---|---|---|
| Gender | 0.5 | 0.501 | 1 | | | |
| Importance | 4.493 | 0.295 | 0.047 ns | 1 | | |
| Proficiency | 1.891 | 0.529 | 0.289 *** | 0.234 *** | 1 | |
| MWDS | 11.692 | 2.433 | −0.257 *** | 0.316 *** | −0.849 *** | 1 |

ns = not significant; *** Correlation is significant at the 0.01 level. MWDS: Mean Weighted Discrepancy Scores.

### 4.2. Importance of Main Production and Agribusiness Management Practices

Table 2 indicates that there is no significant association between gender and the importance of competency statements. This finding is logical since the level of importance of a production skill is not influenced by the gender of the farmers. The detailed analyses of competencies reflect a general tendency of non-significant gender differences in relation to production and agribusiness-related practices (Table 3), except for breeding and rearing ($t$ = −2.326, $p$ = 0.021). However, the effect size analysis for this variable ($d$ = 0.233) reveals a small difference between the gender groups. The ranking pattern of the practices between the gender groups is relatively the same.

**Table 3.** Importance of production and management practices among males and females.

| Competency | Overall (S = 397) | | | Males ($s_{male}$ = 199) | | | Females ($s_{female}$ = 198) | | | t-Value (p-Value) |
|---|---|---|---|---|---|---|---|---|---|---|
| | Mean | SD | Rank | Mean | SD | Rank | Mean | SD | Rank | |
| Disease control | 4.971 | 0.160 | 1 | 4.978 | 0.142 | 1 | 4.965 | 0.176 | 1 | −0.846 (0.398) |
| Marketing and pricing | 4.829 | 0.375 | 2 | 4.862 | 0.328 | 2 | 4.797 | 0.415 | 2 | −1.733 (0.084) |
| Farm business management | 4.713 | 0.382 | 3 | 4.715 | 0.372 | 4 | 4.711 | 0.393 | 3 | −0.103 (0.918) |
| Fodder production and storage | 4.698 | 0.359 | 4 | 4.725 | 0.315 | 3 | 4.670 | 0.397 | 4 | −1.536 (0.125) |
| Pasture management | 4.591 | 0.392 | 5 | 4.616 | 0.360 | 5 | 4.566 | 0.421 | 5 | −1.291 (0.198) |
| Breeding and rearing | 4.275 | 0.444 | 6 | 4.327 | 0.429 | 6 | 4.224 | 0.454 | 6 | −2.326 ** (0.021) |
| Nutritional needs and deficiencies | 4.184 | 0.429 | 7 | 4.208 | 0.381 | 7 | 4.160 | 0.472 | 8 | −1.109 (0.268) |
| Farmers' cooperative management | 4.131 | 0.960 | 8 | 4.099 | 0.984 | 8 | 4.163 | 0.939 | 7 | 0.668 (0.505) |
| Shed and pen construction | 3.979 | 0.694 | 9 | 4.010 | 0.702 | 9 | 3.947 | 0.686 | 9 | −0.906 (0.366) |

*p*-values in parentheses; ** $p < 0.05$.

Disease control obtained the highest rank for both males and female farmers ($M_{Males}$ = 4.978, $M_{Females}$ = 4.965). The high prevalence risk of tick-borne diseases and the threat of the foot and mouth disease outbreaks due to the tropical climatic conditions, have induced strict legally tied disease presentation measures in the country [48]. Farmers are also keen on reporting and addressing diseases through consultation with trained locally deployed veterinary assistants that supply weekly reports to regional disease surveillance centers [2]. This has raised awareness on the importance of livestock disease control among farmers, thus the high ranking.

A distinct pattern is evident within the top five ranked practices. The set of agribusiness management related practices, marketing and pricing ($M_{Males}$ = 4.862, $M_{Female}$ = 4.797) and farm business management ($M_{Males}$ = 4.715, $M_{Females}$ = 4.711), obtained the second-best ranking (overall ranking of 2nd and 3rd, respectively). Agribusiness-orientedness and correct pricing is at the center of the economic growth debate in developing and emerging economies [49]. Organization of production systems and competitiveness in pricing and marketing skills are basic for the transition towards intensive agribusiness-oriented farming. Therefore, the importance of agribusiness-related concepts in farmer training programs is critical for rural and national economic growth.

The set of practices related to feed and feeding obtained 4th and 5th overall ranking, respectively. Considering the pasture-based grazing system in Eswatini, pasture management ($M_{Males}$ = 4.616, $M_{Females}$ = 4.566) is key in the production of high-quality fodder that is necessary for the production of high-quality cattle that fetch high market value [46]. Pasture management must also ensure sufficient forage throughout the year, reducing the cost of supplement feed [45]. This category of production management practices is, therefore, important in increased production of marketable surplus that is essential for the shift towards agribusiness farming. Training on feed and feeding practices is also pivotal in the cost-effective production of cattle according to the unique nutritional needs of the different classes of cattle [47].

Breeding-related concepts are paramount in the selection of high-quality productive breeds and individuals with the herd. Productive breeding stock increases the volume of marketable surplus, while high-quality breeding stock increases growth rate and body conditions. These allow for increased production of high-quality marketable surplus that improves market value. In turn, this creates a market incentive that promotes engagement in agribusiness farming. Therefore, training on breeding-related concepts is highly recommended [32].

The importance of farmer cooperatives as an economic developmental mechanism for developing countries cannot be overemphasized. Cooperatives enhance production, marketing, and mobilization of financial capital required in the production and marketing of marketable surplus [64]. This is a

critical role in the transition towards agribusiness-oriented farming, thus the necessity of training in this aspect.

Practices related to the construction of proper cattle sheds and pens received the least rank (overall intermediate mean of 3.979 on a scale of 5). In the study area, farm structures relate to kraals and crushes that confine cattle overnight. The least importance rank is attributed to the simplicity of the construction of kraals, making use of readily available wood material. However, the shift towards intensive agribusiness in beef cattle farming requires the construction of specialized structures such as water and feed troughs that have specific dimensions. The installation and use of equipment such as weight scales requires training, hence the importance of proper farm structures in training programs.

### 4.3. Proficiency in Production and Agribusiness Management Practices

On the scale of 5, the overall means for all assessed competencies range from 1.193 to 2.775, indicating a low proficiency among the farmers (Table 4). Farmers require vigorous training in cattle production and agribusiness management practices, otherwise, intensive production of marketable surplus will continue to be undermined. This will further undercut increased economic benefit for farmers through intensive agribusiness farming. The detailed analysis for each production and agribusiness practice reveals a general tendency of higher proficiency mean scores for male farmers compared to females, although the difference margins are small. The implication is that male farmers are better able at applying production and related agribusiness management practices compared to their female counterparts.

**Table 4.** Proficiency in production and management practices among beef cattle farmers.

| Competency | Overall ($S = 397$) | | | Males ($s_{male} = 199$) | | | Females ($s_{female} = 198$) | | | t-Value (p-Value) | d-Value |
|---|---|---|---|---|---|---|---|---|---|---|---|
| | Mean | SD | Rank | Mean | SD | Rank | Mean | SD | Rank | | |
| Disease control | 2.775 | 0.969 | 1 | 3.214 | 0.843 | 1 | 2.333 | 0.884 | 1 | −10.162 *** (0.000) | 1.020 |
| Breeding and rearing | 2.541 | 0.625 | 2 | 2.810 | 0.541 | 2 | 2.270 | 0.586 | 2 | −9.543 *** (0.000) | 0.958 |
| Sheds and pen construction | 1.883 | 0.496 | 3 | 2.113 | 0.473 | 3 | 1.652 | 0.403 | 4 | −10.472 *** (0.000) | 1.050 |
| Pasture management | 1.838 | 0.508 | 4 | 2.042 | 0.451 | 4 | 1.633 | 0.481 | 6 | −8.742 *** (0.000) | 0.877 |
| Farm business management | 1.804 | 0.934 | 5 | 1.869 | 0.985 | 6 | 1.739 | 0.877 | 3 | −1.382 (0.168) | 0.139 |
| Fodder production and storage | 1.782 | 0.371 | 6 | 1.916 | 0.333 | 5 | 1.646 | 0.358 | 5 | −7.776 *** (0.000) | 0.780 |
| Marketing and pricing | 1.710 | 0.892 | 7 | 1.829 | 0.951 | 7 | 1.590 | 0.813 | 8 | −2.698 *** (0.007) | 0.271 |
| Cooperative management | 1.658 | 0.845 | 8 | 1.714 | 0.907 | 8 | 1.603 | 0.775 | 7 | −1.309 (0.191) | 0.131 |
| Nutritional needs and deficiencies | 1.193 | 0.417 | 9 | 1.229 | 0.478 | 9 | 1.157 | 0.344 | 9 | −1.745 (0.082) | 0.175 |

p-values in parentheses; *** $p < 0.01$. MWDS: Mean Weighted Discrepancy Scores.

Disease control obtained the highest rank for both groups, but males showed intermediate proficiency ($M = 3.214$, $SD = 0.843$), whereas female farmers showed a low proficiency ($M = 2.333$, $SD = 0.884$). The high ranking for disease control is attributed to the combined effect of the national legal disease preventative measure that enforces prophylactic tick-borne disease control [2] and the keenness among farmers to prevent livestock losses through diseases. The results are in line with Ampaire and Rothschild [11] who identified training need in animal disease management and treatment to improve livestock physical characteristics that are critical for market value. The variable also revealed a large effect size (Cohen's $d = 1.020$), implying that males are significantly very much better than females ($t = −10.162$, $p = 0.000$) at controlling cattle diseases. This comparison implies more training need for females in this regard.

It is worth noting that both male and female farmers reported very low overall proficiency in production and related agribusiness management practices from the third overall ranked practice. Farmers are less competent in the application of skills in construction of farm structures, pasture management, farm business management, fodder production and storage, marketing and pricing, cooperative management, and identifying and dealing with cattle nutritional needs and deficiencies. The results present an urgent need for vigorous farmer training programs since farmers reported low competence in almost all the production and agribusiness-related practices. Lack of technical and scientific knowledge in livestock production is a general problem in developing countries [10], such as in sub-Saharan Africa.

Both males and females, although males are significantly better than females, have low proficiency in breeding and rearing ($M_{Male}$ = 2.810, $M_{Female}$ = 2.270, $t$ = −9.543, $p$ = 0.000, $d$ = 0.958). Similar results were found by Durgga and Subhadra [32] who reported that livestock breeding-related concepts were the most important training need required by female farmers in India. The results reveal a significant difference in relation to competencies in construction of proper sheds and pens ($M_{Male}$ = 2.113, $M_{Female}$ = 1.652, $t$ = −10.472, $p$ = 0.000, $d$ = 1.050), indicating that males are much better than females in this regard.

Furthermore, the statistical assessment for mean difference between gender groups reveals significant differences regarding pasture management ($M_{Males}$ = 2.042, $M_{Females}$ = 1.633, $t$ = −8.842, $p$ = 0.000, $d$ = 0.877), fodder production and storage ($M_{Males}$ = 1.916, $M_{Females}$ = 1.646, $t$ = −7.776, $p$ = 0.000, $d$ = 0.780), and marketing and pricing ($M_{Males}$ = 1.829, $M_{Females}$ = 1.590, $t$ = −2.698, $p$ = 0.007, $d$ = 0.271). Due to the communal grazing system and the use of crop remains to supplement winter grazing, pasture management and fodder production are not common in the study. High-quality fodder production is critically vital for beef cattle farming [46] in this era of climate change that continues to impose adverse weather conditions on grazing-based farming systems. Therefore, advancing farmers' skills and knowledge in fodder production is one of the primary issues of the time [10,45].

The lack of awareness in fodder production has further imposed the least competence in identifying and addressing cattle nutritional needs and deficiencies for both males and females ($M_{Males}$ = 1.229, $M_{Females}$ = 1.157). Knowledge and skills in addressing cattle nutritional needs are necessary in ensuring high livestock market value, thereby creating marketing incentive for agribusiness-oriented farming and market participation. Understanding and addressing the factors that affect cattle nutritional needs, according to the different cattle classes is prime in cost-effect markets and consumer-oriented production [47].

Market imperfection, induced by the lack of an organized cattle market structure [50], has deprived farmers of market price information, hence the low competence levels in cattle marketing and pricing. The insignificant difference between the gender groups regarding cooperative and farm business management is logical since such practices are not embedded in the traditional farming system that serves as the basis for household farm education. Both aspects received low proficiency scores, farm business management ($M_{Males}$ = 1.869, $M_{Females}$ = 1.739) and cooperative management ($M_{Males}$ = 1.714, $M_{Females}$ = 1.603). These concepts are very important in creating market competitiveness [49] and organizing production and in the mobilization of production resources [64] for improved livelihoods and stability [51]

### 4.4. Competency Discrepancies among Beef Cattle Farmers

The mean weighted discrepancy scores for the overall sample range from 7.414 to 15.066 (Table 5). A general tendency of higher mean weighted discrepancy scores for females over males is evident in Table 5. This means that female farmers tend to require more training compared to the male farmers.

Generally, the results indicate that training priority should be awarded to agribusiness-related practices, followed by feed and feeding practices, cattle health concepts, farmer organizational concepts (farmers' cooperative management), construction and maintenance of cattle sheds and pens, and breeding and rearing practices. Agribusiness-related management practices, marketing and pricing

($MWDS_{Male}$ = 14.646, $MWDS_{Female}$ = 15.488), and farm business management ($MWDS_{Male}$ = 13.417, $MWDS_{Female}$ = 14.009), obtained the highest overall ranking (1st and 2nd, respectively). Information dissemination in these concepts is critical in building farmers' competitiveness [49], especially under conditions of market imperfection that deprive smallholder farmers of market price information [50]. Smallholder farmers with reduced market competence often suffer market segregation and fail to penetrate formal markets [52]. This further reduces market participation incentive [28], thereby undermining the shift towards intensive agribusiness farming.

**Table 5.** Mean weighted discrepancy scores among beef cattle farmers.

| Competency | Overall ($S$ = 397) | | | Males ($s_{male}$ = 199) | | | Females ($s_{female}$ = 198) | | | *t*-Value (*p*-Value) | *d*-Value |
|---|---|---|---|---|---|---|---|---|---|---|---|
| | MWDS | SD | Rank | MWDS | SD | Rank | MWDS | SD | Rank | | |
| Marketing and pricing | 15.066 | 4.304 | 1 | 14.646 | 4.547 | 1 | 15.488 | 4.013 | 1 | 1.956 (0.051) | 0.196 |
| Farm business management | 13.712 | 4.382 | 2 | 13.417 | 4.480 | 2 | 14.009 | 4.271 | 3 | −1.383 (0.168) | 0.135 |
| Fodder production and storage | 13.699 | 2.327 | 3 | 13.196 | 2.160 | 3 | 14.204 | 2.385 | 2 | 4.413 *** (0.000) | 0.443 |
| Pasture management | 12.640 | 2.904 | 4 | 11.820 | 2.664 | 5 | 13.464 | 2.909 | 4 | 5.873 *** (0.000) | 0.589 |
| Nutritional needs and deficiencies | 12.513 | 2.535 | 5 | 12.461 | 2.600 | 4 | 12.566 | 2.474 | 6 | 0.413 (0.680) | 0.041 |
| Disease control | 10.920 | 4.928 | 6 | 8.769 | 4.300 | 7 | 13.081 | 4.568 | 5 | 9.685 *** (0.000) | 0.972 |
| Cooperative management | 10.215 | 4.627 | 7 | 9.853 | 4.819 | 6 | 10.578 | 4.409 | 7 | 1.562 (0.119) | 0.157 |
| Sheds and pen construction | 8.338 | 3.160 | 8 | 7.547 | 3.086 | 8 | 9.133 | 3.039 | 8 | 5.156 *** (0.000) | 0.518 |
| Breeding and rearing | 7.414 | 2.643 | 9 | 6.483 | 2.299 | 9 | 8.351 | 2.640 | 9 | 7.519 *** (0.000) | 0.755 |

*p*-values in parentheses; *** $p$ < 0.01. MWDS: Mean Weighted Discrepancy Scores.

Feed and feeding-related practices, fodder production and storage ($MWDS_{Male}$ = 13.196, $MWDS_{Female}$ = 14.204), pasture management ($MWDS_{Male}$ = 11.820, $MWDS_{Female}$ = 13.464, $t$ = 5.873) and identifying nutritional needs and deficiencies ($MWDS_{Male}$ = 12.461, $MWDS_{Female}$ = 12.566), obtained 3rd, 4th and 5th overall ranking, respectively. Kumar, Vimal, Jiji, and Rajkamal [10] found similar results, recommending training need priority on fodder production and preservation for dairy farmers. Statistically significant differences between the gender groups were revealed with respect to fodder production and storage ($t$ = 4.413, $p$ = 0.000, $d$ = 0.443) and pasture management ($t$ = 5.873, $p$ = 0.000, $d$ = 0.589), implying more training need for females than males.

A significant difference is also revealed in relation to disease control ($t$ = 9.685, $p$ = 0.000), reflecting a strong training need for females compared to males ($d$ = 0.972). The variable is ranked 5th for women and 7th for males, further implying greater training need women in this regard. Luqman, Shahbaz, Khan, and Safdar [38] reported similar findings in a study on rural women in livestock management. Practices related to the construction of proper sheds and pens revealed a statistically significant difference between the gender groups ($MWDS_{Male}$ = 7.547, $MWDS_{Female}$ = 9.133, $t$ = 5.156, $p$ = 0.000) with a moderate effect size ($d$ = 0.518).

Although least ranked, breeding and rearing show a statistically significant difference between the gender groups ($t$ = 7.519, $p$ = 0.000) with a moderate effect size ($d$ = 0.755). The significant difference between the gender groups in relation to this practice implies a greater training need for females than males. Under communal livestock management systems, controlled breeding (selective breeding) is not practiced. Cattle herds from different households mix in the grazing lands and mate without control. Hence, the low ranking of this production and management practice. A previous study by Jacob and George [12] reported similar findings, that livestock farmers lack training on scientific knowledge about cattle management practices.

Cooperative management revealed insignificant differences between group means. Cooperatives in Eswatini are faced with numerous challenges that have induced a negative attitude among farmers [53]. However, the desired shift towards an intensive co-operated agribusiness system requires a redress of co-operativism as a mechanism for economic rural development. Hence, the need for training in co-operativism.

### 4.5. Overall Training Needs for Female Beef Cattle Farmers

Table 6 presents an independent t-test comparison for the gender group means to ascertain the overall training needs for female farmers. The results indicate high importance ranking for the competences in cattle production and agribusiness management, $M = 4.493$ on a scale of 5. Generally, the group means reflect small difference margins between the gender groups in all the attributes of assessment (importance, proficiency, and discrepancy). The overall t-test reveals a non-significant mean difference between male and female farmers ($t = -0.933$, $p = 0.352$) with a nugatory effect size ($d = 0.094$), in relation to the importance of production and agribusiness management practices. Importance of any production and management practices does not depend on the gender of the farmer.

**Table 6.** Training need comparison between male and female farmers.

| Variable | Sample ($S = 397$) | | Males ($s_{male} = 199$) | | Females ($s_{female} = 198$) | | $t$-Value ($p$-Value) | $d$-Value |
|---|---|---|---|---|---|---|---|---|
| | **Mean** | **SD** | **Mean** | **SD** | **Mean** | **SD** | | |
| Importance | 4.493 | 0.295 | 4.507 | 0.279 | 4.480 | 0.310 | −0.933 (0.352) | 0.094 |
| Proficiency | 1.891 | 0.529 | 2.044 | 0.528 | 1.738 | 0.484 | −6.004 *** (0.000) | 0.603 |
| Discrepancy | 11.692 | 2.433 | 11.069 | 2.340 | 12.317 | 2.369 | 5.280 *** (0.000) | 0.530 |

$p$-values in parentheses; *** $p < 0.01$.

Based on the Likert-scale used for the analysis, the sample overall mean for proficiency in production and agribusiness-related is very low ($M = 1.891$). This indicates that farmers are incompetent in the application of production and agribusiness-related competencies in beef cattle farming. In addition, the statistical test reveals a significant mean difference between male and female farmers ($t = -6.004$, $p = 0.000$) in relation to farmers' proficiency in production and related agribusiness practices. Although the magnitude of the discrepancy between the group means ($M_{Males} = 2.044$, $M_{Female} = 1.738$) is low, the need for more training for female farmers can never be neglected in developing countries. The moderate proficiency effect size ($d = 0.603$) endorses the claim for more training for female farmers. Therefore, vigorous and pragmatic training programs must be mounted to enhance the knowledge and proficiency levels among beef cattle farmers. Attention must be given to gender- diversity to embrace the special training needs for women, who lag behind males [30]. This is critical for a gender-inclusive progressive shift towards intensive agribusiness that advances rural development and national economic growth.

Regarding the discrepancy between importance and proficiency, the group means divulge small difference margins ($M_{Males} = 11.069$, $M_{Females} = 12.317$). Generally, the results indicate a serious training need for beef cattle farmers in the country. Furthermore, the statistical test reveals a strong significant difference between the group means ($t = 5.280$, $p = 0.000$). This resounds the necessity of more training need for females compared to males, a conclusion that was also made by Jadav, Rani, Mudgal, and Dhamsaniya [13]. The moderate effect ($d = 0.530$) transforms this necessity into an obligation, considering the 63% poverty rate [40] and the above 40% unemployment rate [41] in the study area. Moreover, a report by FAO [37] indicated that women farmers are less privy to training opportunities, yet Luqman, Shahbaz, Khan, and Safdar [38] found that female farmers lack technical and scientific knowledge in livestock production. Therefore, the findings of this study reveal

gender-specific farmer training need to be an indispensable strategic tool for the shift toward intensive agribusiness-orientedness in beef cattle farming. Besides the macro outlook of the agrarian-economic base for Eswatini, women are directly involved in household food security [30,33], deserving more vigorous educational capacitation.

## 5. Conclusions

The importance of production and agribusiness related management practices was found not to be dependent on the farmer's gender. Competency items were perceived to be of high importance (4.493 on a scale of 5) in beef cattle production and the related agribusiness management. However, this study provides an evidence-based statistically significant gender effect on proficiencies and discrepancies in beef cattle production and agribusiness related management practices. Farmers show a very low overall mean proficiency score (1.891 on a scale of 5) implying that farmers are less competent in beef cattle production and agribusiness management. Furthermore, females were found to be less competent than male farmers, thus having a greater need for training.

In the drift towards Vision 2022, a national strategy to achieve first-world status by 2022, it is recommended that the government should mount a rigorous training program for beef cattle farmers. The program can be disseminated through the Livestock Extension Department of the Ministry of Agriculture. Special attention should be given to female farmers who lag behind male farmers. Dissemination of training information should be prioritized as follows: (1) agribusiness concepts, (2) feed and feeding concepts, (3) cattle health concepts, (4) farmer organizational concepts, (5) construction of proper sheds and pens, and (6) cattle breeding and rearing concepts.

**Author Contributions:** Conceptualization, methodology and validation, S.I.D. and W.-C.H.; formal analysis, investigation, resources, data curation, writing—original draft preparation, S.I.D. writing—review and editing, S.I.D. and W.-C.H.; supervision W.-C.H. All authors have read and agreed to the published version of the manuscript.

**Funding:** This research received no external funding.

**Conflicts of Interest:** The authors declare no conflicts of interest.

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
