# Peer review of "Towards Intensive Co-operated Agribusiness: A Gender-Based Comparative Borich Needs Assessment Model Analysis of Beef Cattle Farmers in Eswatini"

_agriculture, doi:10.3390/agriculture10040096_

Round 1

Reviewer 1 Report

The paper presents the issue of the beef cattle farming in Eswatini, highlighting the gender disparity in proficiency due to the socio-economic situation. Some cultural and demographic characteristics of population lead to a lower performance of females in cattle breeding respect to males. The differences, due to the gender, point out as training is a mandatory issue to fill the gap between men and women.

The work is very interesting and well supported by a considerable sample (397 interviews of farmers). Considering that the economic and agricultural situation of this Country is quite unknown from a scientific point of view, this survey is pivotal to understand the scenario and to support some political decisions that can improve the rural livelihood.

Statistical analysis is well developed and illustrated, but there are some results that not meet the correct interpretation of the values. At the bottom of table 3, 4 and 5 an overall t-test was reported and it assesses the difference between the overall means of the males vs females. The unique average that was reported refers to an overall mean that was calculated for all the sample without distinction between males and females. It would be best if you reported the overall means for males and for females because the overall average without distinction is not relative to t-test. In general, regarding the use of the t-test, did you perform some analyses of the data distribution?

Some concepts are repeated numberless times increasing the weight of the speech. In particular in results and discussion section it would be more sliding if the focus concept of the need of training of the women was discussed in a unique part of the text, supported by all the analyses presented.

Below my punctual observations.

Line 23. The average reported, M=11.69, is not a measure of the gap between males and females, but an average value of all the mean weighted discrepancy scores calculated for all items considered in the questionnaires. It is an average of the discrepancy between importance and proficiency (as stated in line 257) without considering the gender of respondents. It would be better to repot only the value of t-test as done in line 22.

Line 46 and 147 Could you provide a different word instead of ‘inculcating’? It doesn’t sound politically corrected.

Table 1: when you report the mean values for continuous variables (age, herd size, experience) it would be better if you report standard deviation too. Did you perform the analysis of the distribution of data?

Table 2: Even if the correlations are highly significant, the absolute values (for gender vs proficiency and MWDS and for Importance vs proficiency and MWDS) are quite low, if we consider that the range of this index is between -1 and 1. The significance is due to the large sample size: when the dimension of sample increases the significance of correlation increases too, even if the value is not too high. I’d keep the results of correlation less than 0.6 with caution, whereas the correlation between Proficiency and MWDS is high and significant.

Table 3,4 and 5: As suggested before, at the bottom of the tables it would be better if the overall means for males and females were reported.

In general, when you comment the result of the overall t-tests (line 331, 354, 383) it would be better if you report only the t-test e p values and not the overall mean (which is indistinct for gender).

Reviewer 2 Report

Well presented results and summaries. but should support evidence with literature. then it can be a great paper. Quite relevant and I enjoyed reading. Used quite high English vocabulary, just need some editing as I have tracked changed.
